# The Important Role of Endothelium and Extracellular Vesicles in the Cellular Mechanism of Aortic Aneurysm Formation

**DOI:** 10.3390/ijms222313157

**Published:** 2021-12-06

**Authors:** Klaudia Mikołajczyk, Dominika Spyt, Wioletta Zielińska, Agnieszka Żuryń, Inaz Faisal, Murtaz Qamar, Piotr Świniarski, Alina Grzanka, Maciej Gagat

**Affiliations:** 1Department of Histology and Embryology, Faculty of Medicine, Collegium Medicum in Bydgoszcz, Nicolaus Copernicus University in Torun, 85-092 Bydgoszcz, Poland; klaudia.mikolajczyk@cm.umk.pl (K.M.); dominika.spyt@cm.umk.pl (D.S.); w.zielinska@cm.umk.pl (W.Z.); azuryn@cm.umk.pl (A.Ż.); inazfaisal.1009@gmail.com (I.F.); murtaz.qamar@gmail.com (M.Q.); agrzanka@cm.umk.pl (A.G.); 2Department of Urology and Andrology, Faculty of Medicine, Collegium Medicum in Bydgoszcz, Nicolaus Copernicus University in Torun, 85-092 Bydgoszcz, Poland; piotr.swiniarski@hotmail.com

**Keywords:** inflammation, aneurysm, atherosclerosis, cell migration, extracellular vesicles

## Abstract

Homeostasis is a fundamental property of biological systems consisting of the ability to maintain a dynamic balance of the environment of biochemical processes. The action of endogenous and exogenous factors can lead to internal balance disorder, which results in the activation of the immune system and the development of inflammatory response. Inflammation determines the disturbances in the structure of the vessel wall, connected with the change in their diameter. These disorders consist of accumulation in the space between the endothelium and the muscle cells of low-density lipoproteins (LDL), resulting in the formation of fatty streaks narrowing the lumen and restricting the blood flow in the area behind the structure. The effect of inflammation may also be pathological dilatation of the vessel wall associated with the development of aneurysms. Described disease entities strongly correlate with the increased migration of immune cells. Recent scientific research indicates the secretion of specific vesicular structures during migration activated by the inflammation. The review focuses on the link between endothelial dysfunction and the inflammatory response and the impact of these processes on the development of disease entities potentially related to the secretion of extracellular vesicles (EVs).

## 1. Introduction

The inflammatory process is a part of the cellular response for severe imbalance in tissue homeostasis. Endothelial cells (ECs) play a crucial role in the activation and course of an inflammatory process [1,2]. The expression of the endothelial adhesion molecules is responsible for the selective recruitment of subpopulations of circulating leukocytes to the inflammation site [3]. Additionally, ECs can produce proinflammatory cytokines, i.e., tumor necrosis factor α (TNF-α), which promotes the secretion of other cytokines, e.g., interleukins 1 and 6 (IL-1, 6). Overexpression of these molecules can result in artery wall remodeling connected with an increase in endothelial permeability, particularly for the circulating LDL in the bloodstream and their accumulation in the inner membrane. Moreover, ECs are characterized by the presence of numerous integrin receptors on their surface, which determines their high affinity for extracellular matrix (ECM) proteins [4,5,6]. Interactions of intercellular binding complexes with the actin cytoskeleton of cells ensure the proper function of vascular endothelium. Based on the studies, it was proposed that regulating the degree of the F-actin protein polymerization could be crucial for maintaining endothelial barrier stability, surface molecules adhesion, and migration potential [7,8]. Further, F-actin cytoskeleton depolymerization and reorganization promotes the changes in ECs proteins expression profile and formation of F-actin stress fibers. It is associated with the loss of coordinated EC migration, which is necessary to maintain the endothelial barrier. The described cause and effect sequence leads to endothelial dysfunction [9,10,11]. The literature data analysis concerning the pathogenesis of the artery vessel diseases has allowed observing secretion of specific vesicle structures in response to point contact between ECs and the surface of the endothelial and extracellular matrix (ECM), which could be related to cell signaling [12]. In addition, different manipulations within ECs involving the activation of expression of regulatory proteins of actin filaments, i.e., tropomyosin 1 (TPM1), indicate the maintenance of connections between ECs under both depolymerization and excessive polymerization of F-actin located in the cortical layer of the cytoplasm [13]. Based on literature data, it can be concluded that the stabilization of F-actin through TPM1 effectively reduces the inflammatory response of human ECs, and thus also the secretion of extracellular vesicles [14,15].

Extensive research on the pathogenesis of cardiovascular diseases at the molecular level gave the basis for a review of literature data on the relationship between aneurysm formation, the development of atherosclerotic lesions, and the secretion of extracellular vesicles accompanying these processes. In the course of the research, secretion of endothelial extracellular vesicles probably could be an attractive therapeutic target in inflammatory cardiovascular diseases. Protein profiling of endothelial extracellular vesicles, assessment at the ultrastructural level, and analysis of fundamental life processes should answer the questions regarding the biological significance of extracellular endothelial vesicles and their possible use as a diagnostic marker or therapeutic molecule in the diagnosis and treatment of the cardiovascular pathologies [16,17].

## 2. The Biology of Aortic Aneurysms

Aneurysms develop within arteries, and their classification is strongly related to their location. The primary function of the arteries is the pulsatile circulation of blood from the heart to the capillaries by rhythmic contractions of cardiac muscles [18,19]. The contraction and relaxation of the heart chambers induce changes in the diameter of the arteries, allowing oxygenated blood to flow to various organs and tissues [20]. The artery wall consists of three layers: the innermost layer known as tunica intima, the middle layer named tunica media, and the outermost layer described as the tunica adventitia (Figure 1) [21]

Tunica intima contains flattened ECs located on a lamina rich in collagen and elastin fibers. The subendothelial layer is part of the supporting tissue composed of myoid cells involved in synthesizing ECM components, i.e., proteoglycans, glycoproteins, and microfiber. Myoid cell layers store lipid substances, which causes thickening of the artery inner layer with age. Due to the accumulation of lipid deposits in the arterial wall, the described process promotes atherosclerotic plaque formation. The other two arterial wall layers consist of elastin, collagen, and smooth muscle cells (SMCs) compounds in the fibers, together with the vasa vasorum present in the tunica adventitia. The elastic fibers present in the structure of the walls of arteries give elasticity to the vessels. In turn, collagen fibers are responsible for maintaining the structural integrity of the vessel wall and resistance to stretching. Replacement elastin fibers by SMCs and inelastic collagen fibers weaken the arterial walls and promote an aneurysm-like structure [22,23,24,25,26].

Aortic aneurysms (AA) take the form of fusiform or circular abnormal bumps in the aortic wall. AA is the most commonly defined as a pathological dilatation of the aortic segment by ≥50% of the initial vessel diameter [27]. The aortic aneurysms may develop in the thoracic (TAA) and the abdominal aorta (AAA), which are also the most common type of aneurysm [28]. Aneurysms can also be classified according to the De Bakey and Stanford standardization based on the level of their dissection. AA is characterized by an asymptomatic disease course, leading to aortic dissection or rupture, which may be lethal. It is estimated that TAA and AAA account for 1–2% of all deaths in Western countries. The leading cause of death, i.e., aneurysm rupture, correlates with an increase in the diameter of a blood vessel. Additionally, calcifications, intra-lumen thrombus, wall stiffness, oxidative stress, and inflammation are other possible causes of aortic aneurysm rupture. According to statistics data, the prevalence of AAAs is within the range of 4–7% in men and 1–2% in women [29,30]. Hereditary occurrence of AAA is in the range of 12–19%. In turn, mortality in the case of AAA is 65–85%. In contrast to the occurrence of AAAs aneurysms, TAAs prevalence ranges from 5 to 10/100.000 people per year. The genetic background accounts for 20% of all TAA aneurysms. The impact of gender on TAAs occurrence indicates that men are more likely to develop this type of aneurysm than women. On the other hand, women are more likely to develop aortic dissection and aneurysm rupture [31]. Despite the rarer incidence, TAA occurrence is associated with poor prognosis. Moreover, the risk of death is 2–3 times higher compared with AAA [31,32,33]. Furthermore, the epidemiology of aneurysms depends on the age of society. In young patients, aneurysms are observed in those who develop simultaneously genetic diseases such as Marfan syndrome or Loeys–Dietz syndrome. On the other hand, in older patients, aneurysms development is associated with arterial hypertension or atherosclerosis [34,35]. It is worth noting that there are differences in the pathophysiology of aneurysms resulting from their location in the aorta [36]. However, studies show common features such as inflammation of the aortic wall, loss of elastin and collagen fibers, and glycosaminoglycans, promoting aneurysms formation.

The level of high technology available allows the selection of methods for imaging aneurysms, such as transesophageal echocardiography (TEE), magnetic resonance imaging (MRI), computed tomography (CT), transthoracic echocardiography (TTE), and chest radiograph (CXR). Furthermore, increasing progress has been made in the treatment of aneurysms. In addition to standard pharmacological and surgical therapy, biomarkers and stem cells are also widely used. It is noteworthy that it is possible to apply genome editing technology based on CRISPR/Cas9 in the targeted therapy, which seems to be a promising future for treating aortic aneurysms [37,38,39,40,41,42,43,44].

## 3. Pathophysiology of Aneurysms

The aorta undergoes constant reconstruction changes resulting from mechanical properties related to the blood vessel structure and its stiffness, as well as from hemodynamic shear stress and peripheral stretching of the vessel, which is the effect of pulsating blood flow in the vessel lumen [45]. In the context of the role of the blood vessel wall in the aneurysm, the artery vessel wall consists of many components such as fibroblasts, EC, SMC, and ECM, which both represent a significant component of the vascular wall. It is a dynamic structure that undergoes continuous physiological remodeling through the degradation and synthesis of new proteins [33,46]. Proteases are involved in this process, mainly metalloproteinases (MMP) family proteins, granzymes, and cathepsins [47]. It has been shown that the hydrolysis of the peptide bonds by cathepsins affects the degradation of ECM, and destruction of collagen and elastin fibers [48]. MMP is one of the most significant families of ECM proteins [49]. An important function of these protein families is the ability to remodel and degrade ECM components. Their action is connected with the formation of reactive oxygen species’ (ROS) products and proinflammatory cytokines. MMPs are involved in cell proliferation, differentiation, and migration, but also cell apoptosis, angiogenesis, and tissue repair. Moreover, the action of MMPs has also been observed in many pathological conditions, including inflammatory processes and consequences of this condition, such as degenerative diseases or neoplasms [50,51].

The activity of the chosen biologically active compounds is compatible with the remodeling of the arterial wall. The consequence of this property is that aortic wall cells contribute to maintaining homeostasis by sensing the local chemical–mechanical environment. In this process, mechanical stimuli are transformed into a biochemical response. This mechanism involves signaling pathways that, depending on the stimulus, influence different cellular responses. As a result, gene expression modulation, migration, proliferation, cell differentiation, and protein synthesis are possible. It is worth drawing attention to the many factors involved in the pathogenesis of aortic aneurysms that regulate the complexity of related signaling pathways and influence the biological function of cells (Figure 2) [52,53].

One pathway that induces aneurysm formation is based on the mitogen-activated protein kinase/extracellular signal-regulated kinases (MAPK/ERK) activity. The mentioned pathway is an intracellular signal transduction pathway that allows signal reception from the receptor, alteration of gene expression, and results in the cellular response. Additionally, it controls the processes related to survival, mobility, differentiation, and apoptosis of cells [54]. Another protein involved in aneurysms progression is p38. This molecule is responsible for producing proinflammatory cytokines and can regulate their expression by modulating transcription factors; therefore, it is involved primarily in immunological and inflammatory processes. Additionally, p38 allows the induction of the expression of adhesion molecules and inducible enzymes [55,56]. One of the factors associated with endothelial dysfunction leading to aneurysm development is ras homolog family member A (RhoA) pathway activation. RhoA is involved in the inhibition of the expression of endothelial nitric oxide synthase (eNOS). Therefore, this pathway is responsible for regulating the tone of blood vessels. In addition, it plays a key role in regulating the contractile phenotype and the cytoskeleton dynamics by affecting SMC, regulation of myosin light chain phosphatase (MLCP), and pericytes. RhoA pathway acts as a mediator of actin reorganization, influencing the process of generating stress F-actin fibers. Moreover, RhoA activation is involved in the monocyte adhesion and transmigration process; therefore, it is defined as the key pathway involved in the regulatory processes related to the permeability of blood vessels. Research has proven that the RhoA pathway regulates the focal adhesion sites and regulates collect actin stress fibers, and it stimulates migration, cell proliferation, and cell division [57,58,59,60,61,62].

The essential and integral part of the aneurysms formation mechanism is transforming growth factor β (TGF-β), responsible for regulating various cellular processes. Cells such as EC, pericytes, and immune cells express the TGF-β receptors. It has been shown that TGF-β induces proliferation, adhesion, migration, production of ECM compounds, and apoptosis but also affects the organization of the cell cytoskeleton, mobility, and invasion. Interestingly, studies carried out in many cell systems have confirmed that TGF-β has a bifunctional effect. Regarding EC, this cytokine shows either a promoting or an inhibitory effect on angiogenesis. Moreover, by inducing growth factors, TGF-β stimulates the proliferation of SMCs. On the other hand, the proliferative effect of SMCs is inhibited at high concentrations of TGF-β. The research results show that the activation of TGF-β in endothelial signaling is involved in the induction of vascular wall inflammation and the atherosclerotic process. Additionally, it induces cytokines, chemokines, receptors, and adhesion molecules on the EC, such as intercellular adhesion molecule 1 (ICAM-1) and vascular cell adhesion molecule 1 (VCAM-1), proving that TGF-β has proinflammatory activity in the EC [63,64,65,66].

## 4. Extracellular Vesicles Secretion Accompanying the Endothelium Destruction Processes

The ability of intercellular communication, i.e., the reaction to signals from the external environment, is necessary for the proper course of the processes of differentiation, cell growth, and maintaining intra-body balance. It can occur through the secretion of active substances in signaling molecules, amino acids, peptides, proteins, nucleotides, retinoids, steroids, and fatty acid derivatives in the form of growth factors and neurotransmitters or cytokines. Signaling molecules are also low-molecular gases that penetrate biological membranes, i.e., nitric oxide (NO) and carbon monoxide (CO) [67,68]. This intercellular signaling may have either local or far-reaching character. In the first case, the signaling molecules move by diffusion, penetrating through the ECM, which results in a fast, but short response from the neighboring cells. The described mechanism does not involve the bloodstream and is referred to as the paracrine route [69,70,71]. The second is called endocrine influence and involves the transport of signal molecules using body fluids and their interaction with distant cells. This is possible due to the expression of specific cell receptors, which initiate physiological, biochemical, and morphological reactions after recognition of the substrate. Intracrine and autocrine interactions are based on the signaling molecules that exert their biological effects inside the cell in which they were produced. The presented types of intercellular communication are based on the presence of extracellular transmitters and constitute a kind of intra- and extracellular secretion [72,73].

Moreover, signaling via surface proteins is distinguished by combining them within different types of cells. A complex of surface protein presented on the surface of one cell, called a signal protein, and complementary effector protein on the surface of the other cell is formed, which determines signal transduction and induction of the intercellular communication pathway [74,75]. Effector proteins may include enzymes, transcription factors, ion channels, as well as cytoskeleton proteins. Most signal molecules are referred to as molecular switches, which means that both their activation and inactivation are induced by a signal that may result from phosphorylation or the presence of energy transporters in the form of Guanosine-5’-triphosphate (GTP) or Guanosine 5’-diphosphate (GDP). Both the strength of the signal sent, and the sensitivity of the cells is a variable value and depends primarily on adaptive mechanisms. In the case of the second parameter, there is a strong dependence on the number and affinity of receptors, which is why the mechanism of strengthening intercellular signaling is extremely important here [76].

The development of research in etiology and the consequences of interactions between cells has allowed the identification of specific molecules called extracellular vesicles (EVs). Erwin Chargaff and Randolph West were first interested in them in 1946 and described them as “procoagulant particles derived from platelets.” In the 1970s, Peter Wolf enriched the knowledge of ‘microcells’ in new facts. He described them as small structures resembling “platelet dust,” and the results of subsequent studies confirmed the diversity of these molecules and their presence in blood, urine, saliva, semen, and amniotic fluid [77,78,79]. EVs are a heterogeneous population of membrane vesicles released by prokaryotic and eukaryotic organisms, both in vivo and in vitro. The vesicles size is varied and ranges from 30 nm to 5 μm. Although EVs have a similar structure, there are major differences regarding the mechanism of their formation and physicochemical properties [80,81,82,83]. Apart from the density, isolation methods, or the presence of specific markers, the size and diameter of the vesicles are the most common EVs classification criteria. According to them, membrane microstructures can be divided into exosomes, exosome-like vesicles, ectosomes, and apoptotic bodies. The secretion of these molecules occurs due to membrane budding during processes such as exocytosis or apoptosis. EVs have an extremely important role in intercellular communication due to the ability to transfer information between different cells without requiring direct contact [84,85,86]. It is possible due to their content, including proteins, messenger ribonucleic acid (mRNA), and microRNA (miRNA) [87]. Due to the internal composition of EVs and their participation in intercellular communication, they have become an attractive research target for many scientists. This allowed us to observe the important role of the described signaling molecules in physiological processes and conditions accompanied by a disturbance of internal homeostasis of both individual cells and whole organisms [87,88]. It allowed concluding that EVs may act as molecular markers, which may be helpful in the diagnosis of serious diseases at the early stages of their development. Due to the importance of intercellular communication in ontogenesis, organogenesis, and pathophysiology, extensive research in the field of cellular communication led to the identification of extracellular vesicles called microsomes. Their biogenesis and secretion are directly dependent on the migration and polymerization of actin; hence the phenomenon was called microcytosis. This newly identified secretory process can play an important role in intercellular communication during physiological and pathophysiological processes. The mechanism of initiation and the course of migration depends on many complex processes, among which migration is the most important. It plays a key role in embryonic development but is also responsible for the movement of immune cells, wound healing, and tissue regeneration [80,81,82,83,84,85,86]. Cell movement is also a pivotal factor during the formation of tumor metastases and in the course of inflammation. The ability to migrate is determined by the type of tissue from which the cells originate. In these cases, cell movement is necessary during the formation of germ layers and the development of organs. Nervous crest cells, i.e., multipotent neuroectodermal cells, move long distances in the embryo, settle in different places, creating, among others, nerve ganglia or ciliary body. A similar situation is observed among cells, i.e., leukocytes, for which movement is an essential element of functioning.

Knowledge of the mechanisms responsible for cell movement is particularly important for modern medicine because it can be used to study the etiology of various physiological and pathophysiological processes. The essence of migration is the active movement of cells, regulated by signaling pathways associated with information molecules, protein signal transducers, or calcium ions [87,88,89]. Initially, research in the field of cell migration was conducted on unicellular amoeboid organisms of the *Dictyostelium Discoedium* species [90]. However, the identification of similar sets of protein markers allowed to expand research on organisms of higher taxonomic classes. It has become extremely important primarily due to the opportunity to learn about the mechanisms of the processes dependent on each other and the interaction between cells of various types depending on the phenomenon of migration. Types of cell movement are defined based on such parameters as cell adhesion, protease activity, the strength of intercellular interactions, and the polarity and organization of the cell cytoskeleton. Due to the number of migrating cells, individual and collective migration may be distinguished [91,92,93,94].

Cell migration is a multi-stage process consisting of polarization, adhesion, the formation of cellular projections, and directional movement [95]. The first one is the effect of creating functionally and morphologically differentiated poles of the cell, i.e., in the front and posterior surface because of the potential difference between the outer and inner surface of the cell membrane. This results in a gradient distribution of signal and regulatory proteins responsible for intracellular processes associated with cytoskeleton remodeling. The consequence of the polarization is extension of the frontal zone of the cell, after which the phenomenon of adhesion and proteolysis of the ECM is observed [96]. This induces rearrangement of the cytoskeleton by, among others, the action of cofilin, gelsolin, and the Actin-related protein 2/3 (Arp2/3) protein complex [97,98,99,100]. The effect of these processes is the shrinkage of the posterior zone of the cell and break of contact with the surface, which leads to directional displacement. All phenomena associated with the change in the architecture of actin filaments, including polymerization, depolymerization, shrinkage of microfilament networks with the participation of motor proteins, or the movement of the contractile layer, are referred to as cytoskeleton dynamics, which is directly related to the formation of migrasomes.

The basis for identifying migrasomes in the presence of structures with a diameter of 0.5–3 μm tightly fixed at the ends or cuts of actin retraction fibers is through their secretion which occurs during a process called migracytosis. The microstructures detach from the retraction fibers and are released into the extracellular space or directly taken up by surrounding cells. The described vesicles contain inside from less than 10 to 300 smaller follicular structures with a diameter of 50–100 nm, which is why they are often referred to as pomegranate-like structures (PLCs) [101]. It is supposed that migrasomes reflect the composition of the cells from which they are secreted due to the active transport of cytosolic content from the parent cell to the vesicles. During their duration, the content of the parent cells translocates to the migrasome, after which the extracellular vesicle formation rate decreases. Migrasomes, due to their structure, are multivesicular bodies (MVB) like structures, which through fusion with the membrane of the parent cell, contribute to the formation of exosomes. An important feature that makes migrasomes like the MVBs and intraluminal vesicles (ILVs) that build them, is the expression of surface proteins, which in this case are tetraspanin-4 and tetraspanin-7. The similarity of these two types of extracellular vesicles may indicate their common origin, which has become the basis for the conclusion that migrasomes arise because of MVBs migration to retraction fibers. According to this hypothesis, migrasomes should have a bilayer cell membrane, in which the outer layer comes from retraction fibers, while the inner one from MVBs contains surface markers characteristic for MVBs, i.e., LAMP1 (lysosomal-associated membrane protein 1). The results of scientific research indicate, however, strong discrepancies in the described issues, which include, among others, the presence of a single-layer cell membrane or the lack of LAMP1 expression, which made the hypothesis of MVBs as precursors of migrasomes contradictory. Furthermore, other research has shown large differences between extracellular follicular secretion through exocytosis and migracytosis, which mainly concern how they are secreted. In exocytosis, exosomes are released from the cells by MVB fusion to the plasma membrane. At the same time, migracytosis involves the translocation of the cytoplasmic material into vesicles located at the ends of the retraction fibers, resulting in the release of migrasomes [101,102]. Migracytosis and exocytosis are different processes because of their course and the diversity within the markers assigned to the signal molecules secreted during these processes. In the case of migrasomes, this group is represented by proteins such as N-deacetylase/N-sulfotransferase (NDST1), phosphatidylinositol glycan anchor biosynthesis class K (PIGK), and carboxypeptidase Q (CPQ) that are not expressed by other types of signaling molecules. One of the most important functions of these proteins is participation in the inflammatory reaction. On the surface of migrasomes, the presence of transmembrane receptor proteins that connect cells with the ECM, i.e., integrin α5 and β1b, was also noted. They constitute a group of compounds responsible for the adhesion of migrasomes to the place where they arise, whereby the overexpression of these proteins is observed at the contact point between migrasome and retraction fiber [103,104]. The structure of integrins present on the surface of vesicles is characterized by the presence of the pleckstrin homology (PH) domain. The phosphorylation of these domains affects, among others, increased secretion of inflammatory cytokines. The association of migrasomes with the inflammatory response is also seen in the high affinity of these vesicles to proinflammatory proteins, among which fibronectin can be distinguished. Simultaneously, migrasomes can be assigned as a new type of cell organelles.

They perform specific functions and are surrounded by a cell membrane composed mainly of cholesterol, which allows them to be defined as specialized cell subunits. Based on the collected information, it is possible to determine migrasomes as a particularly attractive type of signaling vesicles in the context of diseases initiated by the endothelial inflammatory response due to the high rate of migration of immune cells during the induction of atherosclerosis. Identifying the protein profile of the described EVs, analysis of their biosynthesis, and a broader understanding of the mechanism of microsomes formation may contribute to improving the profile of diagnostics and therapy of cardiovascular diseases [101,102,103,104,105,106]. However, the most frequently described extracellular vesicles are exosomes. This type of extracellular vesicles plays a pivotal role in vascular remodeling which could be an interesting biomarker in the clinic. They can be widely used due to their low toxicity and immunogenicity, biocompatibility, and biological barrier permeability. Enrichment of the content of exosomes in order to use them for targeted therapies can be achieved through co-transfection into the donor cells with plasmid or virus encoding the precursor miRNAs or small interfering RNA (siRNA), electroporation of synthetic miRNAs or siRNAs and also transient transfection of miRNAs using commercial transfection reagents. EVs play an important role in cell-to-cell communication within the disease microenvironment. As some profiling studies have shown, miRNAs encapsulated in secreted EVs have been identified in the extracellular space. The rich content of miRNAs is extremely important in terms of intercellular communication. EVs can serve as valuable biomarkers of various diseases due to the miRNA content. miRNA is a single-stranded RNA molecule on average containing 22 nucleotides [107,108]. miRNA is responsible for regulating gene expression at the post-transcriptional level. The mechanism of action of miRNA is based on interfering with transcription or inhibiting translation. miRNAs are involved in various biological processes, such as differentiation, proliferation, metabolism, and cell death. The loading of miRNAs into the extracellular vesicles protects miRNA from acting of the enzymatic factors such as RNase, but also physics conditions, for example, extreme temperatures or changing pH values. Due to the different molecular mechanisms of the aneurysm formation, growth, and rupture, various types of miRNA patterns are observed. Recent studies indicate that miR-320b, miR-133a, miR-223, and miR-143/145 included in the plasma circulated exosomes could be significant predictors and diagnostic markers of vascular inflammation and atherosclerosis. miR-133a act an important role as a diagnostic marker of myocardial infarction. miR-133a is connected with altered aortic fibroblast phenotype which is observed during aneurysm progression, which weakness the artery wall [109]. The described phenomenon is particularly observed in the TAA course which is a result of dysregulated remodeling of the extracellular matrix on the vascular level and may be connected with altered resident cellular phenotype. miRNA-133a is commonly reduced in TAA clinical specimens and plays an inhibitory role in the pathological phenotypic regulation of switch of vascular cells. Accordingly, it is believed that miR-133a replacement attenuates the development of TAA. These results suggest that stable alterations in aortic fibroblasts may be connected with pathological extracellular matrix remodeling, and regulation by miR-133a may lead to the novel therapeutic strategy [110,111]. Moreover, the growing number of studies show that EV secreted by different types of stem cells transfer their bioactive content to other, more mature cells and thus participate in tissue repair, including the ischemic heart muscle. Due to the fact that different fractions of stem cells secrete vesicles with different molecular compositions, the conducted research is focusing on understanding the biological and regenerative activity of such fractions, including EVs secreted by induced pluripotent stem cells (iPSCs) and mesenchymal stem cells (MSCs) which constitute the fraction of stem cells isolated from mature tissues including bone marrow, adipose tissue, and umbilical cord. Research shows that vesicles secreted by iPSCs (iPSC-EVs), as well as MSCs (MSC-EVs), carry a range of bioactive molecules reflecting the molecular composition of the cells that secrete them, including proteins, mRNA molecules, and small regulatory RNA molecules—the so-called miRNAs which play an important role in regulating a number of genes in cells. Interestingly, the molecular content of the vesicles can be transferred to the cells of the heart and vessels, and thus affect their functions in vitro and after transplantation in vivo. Despite the fact that both iPSCs-EVs and MSC-EVs fractions of EVs have been observed to show proregenerative activity in the heart tissue, they carry a different molecular content, including whole groups of miRNAs, which may differently regulate the functions of different types of cells [112,113,114].

Alteration of the miRNA profile contributing to atherogenesis could serve as an early biomarker in cardiovascular diseases. Recent studies have shown that among smokers, increased level of the circulating leukocyte-derived PMVs (lPMV) and miR-29b is observed. This suggests that a high level of the miR-29b can lead to oxidative stress and induction of inflammatory response. It is connected with the increased production of membrane microvesicles (MVs) by the cigarette smoke-exposed neutrophils. The secreting mechanisms of the microvesicles and the presence of the miR-29b expression is connected with enzymatically active transmembrane ADAM proteases. The high level of the ADAM10 and ADAM17 accompanying to microvesicles secreted mechanism by the neutrophils on exposure to cigarette smoke indicates the molecular mechanism underlying the dramatically elevated risk of abdominal aortic aneurysm. According to the current reports, among the miRNA connected with the aneurysm development, miR-29a, miR-29b, and miR-29c also have been distinguished. The increased level of the miR-29a family is connected with the reduction of the expression of genes encoding extracellular matrix proteins such as fibrillin 1, elastin, and collagen types I and III. Elevated miR-29 expression in the aortic tissue is connected with ECM degradation, contributing to the weakened aortic wall. It makes the aortic wall more susceptible to the aneurysm development. Research proves that the use of the miR-29 inhibitor likewise prevents aneurysm development by reducing the synthesis of the tissue inhibitor of metalloproteinase 3 (TIMP3). This glycoprotein binds to the catalytic domain of MMPs and hydrolyzes the ECM components. Expression of the miR-205 inhibits the level of the reversion-inducing cysteine-rich protein with kazal motifs (RECK) synthesis which inhibits metalloproteinases activity. Research also indicates another significant type of miRNA involved in the progression of aneurysm. It is miR-24, the low level of which can be seen in samples obtained from patients with developed changes in the structure of the arterial wall. Expression of miR-24 is associated with the downregulation of the synthesis of chitinase 3-like protein 1 (CHI-3L1). The low miR-24 level has been connected with proinflammatory interleukin-6 (IL-6) overexpression, which results in an increase in CHI3L1 expression. The CHI3L1 and IL-6 cooperation mechanism contribute to the increase in the production of proinflammatory factors such as intercellular adhesion molecule 1 (ICAM1), vascular cell adhesion molecule (VCAM1), and P-selectin (P-SELP) by the endothelial cells. The secretion of mentioned molecules leads to the binding of leukocytes to the surface of the endothelium. Expression of the CHI3L1 and IL-6 is also connected with the secretion of the interleukin 8 (IL-8) and monocyte chemoattractant protein 1 (MCP1). This phenomenon also contributes to the increased migration of smooth myocytes. Another marker of aneurysm development is miR-29c-3p. Current reports indicate a correlation between miR-29c-3p and the size of the aneurysm that has been observed. Among the genes regulated by miR-29c-3p, there are genes encoding vascular endothelial growth factor-A (VEGF A), collagen type IV, phosphatase, and tensin homolog (PTEN), which contribute to aneurysm formation and could inhibit endothelial cells proliferation. The current reports implicated that elevated miR-29c-3p is a significant biomarker for aneurysm as the downregulation of the gene encoding vascular wall matrix components leads to greater susceptibility to cardiovascular diseases [115,116,117,118,119].

Aneurysms are characterized by the remodeling of the extracellular matrix and inflammatory response. One of the inflammatory protein families capable of degrading extracellular matrix proteins in arteries is the disintegrin and metalloproteinase with thrombospondin 1 motifs (ADAMTS1). The ADAMTS family of proteins comprises 20 members, where ADAMTS-1 and ADAMTS-4 are particularly well studied. ADAMTS-1 and ADAMTS-4 cause tissue destruction and have been connected with vascular diseases, such as atherosclerosis. Research indicates that an increased expression of the majority of the investigated ADAMTS, especially ADAMTS-1 and ADAMTS-4 members on mRNA level, has a large share in the development of aneurysms associated with the infiltration of inflammatory cells and the destruction of the aortic walls. ADAMTS-4 promotes ECM turnover by digesting proteoglycans including biglycan, aggrecan, and decorin or glycoproteins such as fibronectin. Moreover, studies have shown that ADAMTS-1 inhibits cell proliferation by binding and sequestering growth factors, among which vascular endothelial growth factor and fibroblast growth factor stand out. On the other hand, ADAMTS-1 functions in ECM remodeling by digesting versican and tissue factor pathway inhibitor-2. Versican is a chondroitin sulfate proteoglycan. This is the key ingredient of the ECM. It plays the pivotal role in intercellular signaling and in connecting cells with the extracellular matrix. This suggests that the elevated ADAMTS-1 and ADAMTS-4 protein levels could be partially responsible for the increased versican degradation. Additionally, versican can be cleaved by matrix metalloproteinases, so the degradation products in the destructed aortas may have been the collective result of cleavage by ADAMTS proteins, matrix metalloproteinases, and other extracellular proteases. Moreover, recent studies showed that ADAMTS-4 can also break down matrilin-3, alpha-2-macroglobulin, cartilage oligomeric matrix protein, and fibromodulin. Thus, ADAMTS proteins may promote tissue destruction and disease formation through different mechanisms. The increased level of the ADAMTS protein is revealed by profound destruction of the elastic lamellae, rich macrophage accumulation in the adventitia, increased neutrophil values in peripheral blood, and significantly increased expression of inflammatory factors in the early level of aneurysm. This directly indicates that high ADAMTS proteins expression in the progression of inflammation can contribute to the development of pathological changes in the arterial vessels. The importance of ADAMTS proteins in the development and progression of aneurysms is also confirmed by reports indicating that ADAMTS1 deficiency reduces the formation and rupture of aneurysms. Low ADAMTS1 expression inhibits neutrophil and macrophage infiltration by inhibiting the level of proinflammatory cytokines and macrophage migration in the early stage of aneurysm formation, indicating ADAMTS proteins regulation of the inflammatory response, which may be a new therapeutic target in the treatment of aneurysms [120,121].

## 5. Conclusions

Current literature reports implicate that vascular endothelial dysfunction underlies the development of many cardiovascular diseases. Inflammation is distinguished among the main causes of endothelial dysfunction and is accompanied by the secretion of extracellular vesicles into the bloodstream. Numerous studies highlight the role of extracellular vesicles in intercellular communication in vascular remodeling. Recent studies indicate that the content of extracellular vesicles is potentially significant to therapeutic use due to their content. As an alternative carrier, extracellular vesicles internalize cargo based on the pathological and physiological status, inducing various content-depending effects, subsequently. Due to the presence of specific proteins on the surface of extracellular vesicles, they can act as a biomarker of inflammation, and thus also of endothelial dysfunction and the development of cardiovascular diseases. Extracellular vesicles, due to their transport role, can be used for various manipulations due to the transported content. However, there have been few attempts to increase the cargo or modify it for personalized therapies. Despite recent advances, there are still many issues to be resolved regarding the mechanisms of extracellular vesicles biology, such as formation, release, internalization, and their connection with the inflammatory response. Moreover, despite being an alternative for the prognostic and therapeutic approaches in vascular diseases, extracellular vesicle-based therapies are remained to be elucidated and require further studies before clinical applications.

## Figures and Tables

**Figure 1 ijms-22-13157-f001:**
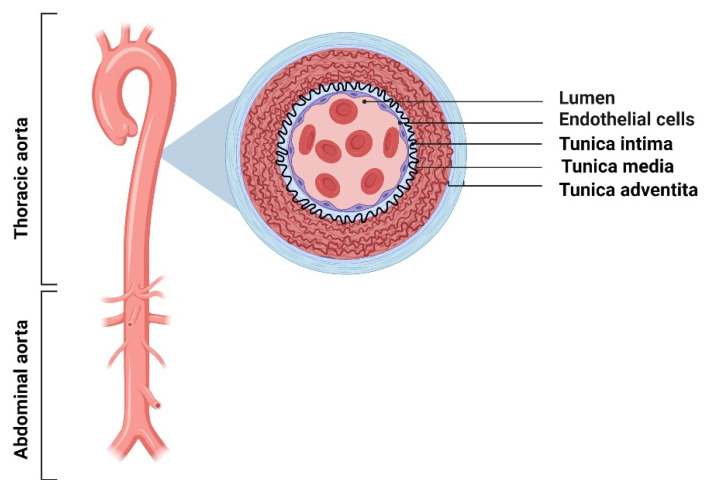
Structure of the layered wall of the human arterial vessel [21,22,23,24,25,26].

**Figure 2 ijms-22-13157-f002:**
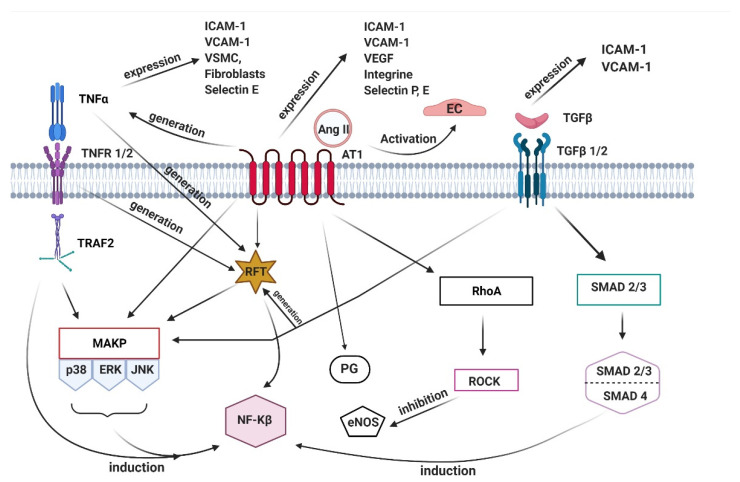
The molecular pathways of the aneurysms formation. One of the inducers of inflammation is angiotensin II (Ang II). It is a peptide hormone, which promotes a proinflammatory, profibrotic, and proliferative effect. This activates EC and induces expression of vascular endothelial growth factor (VEGF), intercellular adhesion molecule 1 (ICAM-1), vascular cell adhesion molecule 1 (VCAM-1), selectins, and integrins. Studies have shown that Ang II, by stimulating the synthesis of prostaglandin (PG) and ROS, modulates microvascular permeability. Consequently, Ang II is involved in the vasodilation mechanism and thus plays an important role in blood pressure regulation. RhoA pathway activation is responsible for barrier dysfunction in the vascular EC (Yao et al., 2010). Moreover, it has been shown that it is involved in the inhibition of the expression of endothelial nitric oxide synthase (eNOS). Interestingly, the binding of AngII to its AT1 receptor in arteries and arterioles induces the production and release of TNFα.

## Data Availability

Not applicable.

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
