# Peer review of "The Important Role of Endothelium and Extracellular Vesicles in the Cellular Mechanism of Aortic Aneurysm Formation"

_ijms, 2021, doi:10.3390/ijms222313157_

Round 1

Reviewer 1 Report

Related to my previous review, authors have followed the recommendations and the conclusion have been reestructured 

Reviewer 2 Report

no further comments

This manuscript is a resubmission of an earlier submission. The following is a list of the peer review reports and author responses from that submission.

Round 1

Reviewer 1 Report

Both Introduction and Conclusion seem to be too long, please revise this sections and attempt to prepare it more condensed. It would be interesting to include the most common clinical stratification (DeBakey and Stanford with short description of clinical implications)

Author Response

Thank you very much for your summary and affirmation of our manuscript. Thank you also for giving us the possibility to improve our manuscript according to your valuable comments.

Point 1: Both Introduction and Conclusion seem to be too long, please revise this sections and attempt to prepare it more condensed.

A: Thank you for your kind advice. According to the suggested, both the introduction and the conclusion have been condensed.
In the case of the introduction, we have removed the sentences containing information on the relationship between inflammation and the development of atherosclerotic plaque, as it is redundant information in relation to the review draft. Revised conclusion is a summary of the issues presented in the review.

Point 2: It would be interesting to include the most common clinical stratification (DeBakey and Stanford with short description of clinical implications).

A: Thank you for your kind advice. We provided the commentary in the chapter "The biology of aortic aneurysms" on classification.

Reviewer 2 Report

 This is an interesting and novel review that provides new insights into the pathogenesis of aortic aneurysm. It is well written and figures are quite illustrative. It opens new horizons for future research , contributing to improve the diagnosis and treatment of aortic diseases. The conclusion is a bit long, and it should be restructured.

Author Response

Thank you very much for your summary and affirmation of our manuscript. Thank you also for giving us the possibility to improve our manuscript according to your valuable comments.

Point 1: The conclusion is a bit long, and it should be restructured.

A: Thank you for your kind opinion on our review. Of course, we made the suggested change, so that the conclusion is a condensed summary of the issues raised in the paper. 

Reviewer 3 Report

Type of manuscript: Review

Manuscript ID: ijms-1425968

Title:  The important role of endothelium and extracellular vesicles in the cellular mechanism of aortic aneurysm formation

Authors: Klaudia Mikołajczyk 1, Dominika Spyt 1, Wioletta Zielińska 1, Agnieszka Żuryń 1, Inaz Faisal 1, Murtaz Qamar 1, 4 Piotr Świniarski 2, Alina Grzanka 1 and Maciej Gagat

In this review Mikołajczyk and colleagues try to summarize the important works related with endothelium function, and extracellular vesicles in aortic aneurysm formation, however the information of the main text does not reflect the title.

In general, the review is poor.

-They did not describe the important pathways in the aneurysm formation, they show them in the figure 2, however they do not talk about at less the signaling pathways that appear in figure 2 as AngII, eNOS that are ones of the main signaling pathways.

-Moreover, there is not almost information about types of extracellular vesicle and the importance of the extracellular vesicles in the aneurysm formation, (mice model described, extracellular vesicles or exosomes profile in human serum of aneurysm patients, extracellular vesicles content and importance in cellular communication in aneurysm, (microRNA etc....).

- The conclusion part is not written as a summary of the review, instead is describing information about the pathophysiology of the aneurysm that should be in the introduction.

-Moreover, a clinical relevance section by the used of extracellular vesicles in the diagnosis of the aneurysm pathology is lack.

Author Response

Thank you very much for your summary and affirmation of our manuscript. Thank you also for giving us the possibility to improve our manuscript according to your valuable comments.

Point 1: They did not describe the important pathways in the aneurysm formation, they show them in the figure 2, however they do not talk about at less the signaling pathways that appear in figure 2 as AngII, eNOS that are ones of the main signaling pathways.

A: Thank you for your thoughtful and thorough review. Your input has been invaluable to make our manuscript more balanced. We have added a figure description containing the key elements of the aneurysm formation pathway. The description is a synthetic introduction to the interaction of peptide hormones, enzymes, adhesive molecules and pro-inflammatory cytokines. The development of this issue could be an excellent topic for further discussion.

Point 2: Moreover, there is not almost information about types of extracellular vesicle and the importance of the extracellular vesicles in the aneurysm formation, (mice model described, extracellular vesicles or exosomes profile in human serum of aneurysm patients, extracellular vesicles content and importance in cellular communication in aneurysm, (microRNA etc....).
Moreover, a clinical relevance section by the used of extracellular vesicles in the diagnosis of the aneurysm pathology is lack.

A: Thank you for these suggestions. We referred to both remarks simultaneously, extending the content of the review to include the potential use of different types of extracellular vesicles in the clinic. The widely described fraction of migrasomes is related to the significant role of this type of extracellular vesicles during inflammation, which is one of the key elements of the review.

Point 3: The conclusion part is not written as a summary of the review, instead is describing information about the pathophysiology of the aneurysm that should be in the introduction.

A: Thank you for these observations. We corrected the conclusion section.

Round 2

Reviewer 1 Report

no further comments

Reviewer 3 Report

I don not believe that the manuscript has been sufficiently improved to warrant publication in IJMS. The information in the main text still does not reflect the review title.

Poor information about the relationship between endothelial extracellular vesicles in aneurysm formation, and works related with that (mice model described, extracellular vesicles or exosomes profile in human serum of aneurysm patients, extracellular vesicles content and importance in cellular communication in aneurysm, (microRNA etc....).